**Data Availability Statement:** All relevant data are within the paper and its Supporting information files. The sequences generated in this study were

# Heterogeneous taxonomic resolution of cytochrome *b* gene identification of bats from Argentina: Implications for field studies

**Diego A. Caraballo**[1]*, **María E. Montani**[2,3,4], **Leila M. Martínez**[5], **Leandro R. Antoniazzi**[6], **Tomás C. Sambrana**[7], **Camilo Fernández**[6], **Daniel M. Cisterna**[5], **Fernando J. Beltrán**[1], **Valeria C. Colombo**[5,6]

1 Instituto de Zoonosis Luis Pasteur, Ciudad Autónoma de Buenos Aires, Argentina, 2 Museo Provincial de Ciencias Naturales "Dr. Ángel Gallardo", Rosario, Santa Fe, Argentina, 3 Programa de Investigaciones de Biodiversidad Argentina (PIDBA), Facultad de Ciencias Naturales e Instituto Miguel Lillo, Universidad Nacional de Tucumán, San Miguel de Tucumán, Tucumán, Argentina, 4 Programa de Conservación de los Murciélagos de Argentina (PCMA), San Miguel de Tucumán, Tucumán, Argentina, 5 Servicio de Neurovirosis, Instituto Nacional de Enfermedades Infecciosas, Administración Nacional de Laboratorios e Institutos de Salud (ANLIS) "Dr. Carlos G. Malbrán", Ciudad Autónoma de Buenos Aires, Argentina, 6 Laboratorio de Ecología de Enfermedades (LEcEn), Instituto de Ciencias Veterinarias del Litoral (ICiVet-Litoral), Universidad Nacional del Litoral (UNL) - Consejo Nacional de Investigaciones Científicas y Técnicas (CONICET), Esperanza, Santa Fe, Argentina, 7 Departamento de Zoonosis, Laboratorio Central de Referencia, Dirección de Promoción y Prevención, Ministerio de Salud de la provincia de Santa Fe, Ciudad de Santa Fe, Santa Fe, Argentina

* diego7caraballo@gmail.com

## Abstract

Bats are among the most diverse, widespread, and abundant mammals. In Argentina, 67 species of bats have been recorded, belonging to 5 families and 29 genera. These high levels of biodiversity are likely to complicate identification at fieldwork, especially between closely related species, where external morphology-based approaches are the only immediate means for *a priori* species assignment. The use of molecular markers can enhance species identification, and acquires particular relevance in capture-release studies. In this study, we discuss the extent of the use of the mitochondrial cytochrome *b* gene for species identification, comparing external morphology identification with a molecular phylogenetic classification based on this marker, under the light of current bat systematics. We analyzed 33 samples collected in an eco-epidemiological survey in the province of Santa Fe (Argentina). We further sequenced 27 museum vouchers to test the accuracy of cytochrome *b* -based phylogenies in taxonomic identification of bats occurring in the Pampean/Chacoan regions of Argentina. The cytochrome *b* gene was successfully amplified in all Molossid and Vespertilionid species except for *Eptesicus*, for which we designed a new reverse primer. The resulting Bayesian phylogeny was congruent with current systematics. Cytochrome *b* proved useful for species-level delimitation in non-conflicting genera (*Eumops*, *Dasypterus*, *Molossops*) and has infrageneric resolution in more complex lineages (*Eptesicus*, *Myotis*, *Molossus*). We discuss four sources of incongruence that may act separately or in combination: 1) molecular processes, 2) biology, 3) limitations in identification, and 4) errors in the current taxonomy. The present study confirms the general applicability of cytochrome

deposited in Genbank under Accession Numbers MT262814 - MT262873.

**Funding:** This work was supported by the Instituto de Zoonosis Luis Pasteur (DAC, FJB), the Instituto Nacional de Enfermedades Infecciosas, Administración Nacional de Laboratorios e Institutos de Salud (ANLIS) "Dr. Carlos G. Malbrán" (DMC), the Curso de Acción para la Investigación y Desarrollo (CAI + D) Orientado 2016 (Res. C.S. N˚ 632/17), Universidad Nacional del Litoral, Agencia Santafesina de Ciencia, Tecnología e Innovación (ASACTEL), Gobierno de la Provincia de Santa Fe (Código IO-2017-00068) (MEM), the Museo Provincial de Ciencias Naturales "Dr. Ángel Gallardo", Ministerio de Innovación y Cultura de Santa Fe (MEM), and the Sociedad Argentina para el Estudio de los Mamíferos (SAREM)(DAC). The funders had no role in study design, data collection and analysis, decision to publish, or preparation of the manuscript.

**Competing interests:** The authors have declared that no competing interests exist.

*b*-based phylogenies in eco-epidemiological studies, but its resolution and reliability depend mainly, but not solely, on the level of genetic differentiation within each bat genus.

## Introduction

Bats (order Chiroptera) include more than 1400 species representing 20% of total mammal diversity [1]. These are among the most widespread and abundant mammals and have the unique capacity of flight. Bats are crucial for the sustainability of many of the world's ecosystems due to their role as massive pollinators, top-down regulators of insect populations, and seed spreaders [2]. However, at least 16% of bat species are threatened [3].

Species identification is not trivial among such a diverse taxonomic group. Traditionally, bats have been identified based on morphological characters or biometric measurements [4, 5], and subsequently by echolocation acoustic analysis [6]. With the advent of molecular techniques, the number of genetic lineages among bats has increased significantly, leading to the identification of many cryptic species, revealing relatively low levels of morphological, biometric, or acoustic differentiation among these mammals [7–10]. Morphologically cryptic species have also been identified by their patterns of echolocation calls and differences in habitat use [11]. Conversely, there are also examples of morphologically distinct species that depict very low genetic distances [12] or share ancestral genetic polymorphisms [13] and, thus, fail to produce reciprocally monophyletic groups when studied on the basis of a limited number of genes.

In Argentina, 67 species of bats have been recorded, belonging to 5 families (Emballonuridae, Noctilionidae, Phyllostomidae, Molossidae, and Vespertilionidae) and 29 genera [14]. These high levels of bat biodiversity are likely to complicate species identification at fieldwork, especially between closely related species, where external morphology-based approaches are the only immediate means for *a priori* species assignment. Although identification keys based on external characters and cranial measurements are available for Argentinian bats [15, 16] the proper use of these keys requires substantial training and experience.

Non-lethal tissue collection methods for subsequent DNA extraction (wing membrane, tail membrane, or tail tips) [17, 18], can enhance species identification in bat capture-release studies and allow for additional research such as phylogeographic analysis [19]. The main pro of this sampling strategy is to affect natural populations as less as possible, acquiring particular relevance in the study of threatened species. Nevertheless, this approach has inherent disadvantages: subsequent studies of viral pathogens, parasites, isotopic analyses, to list a few, are not possible when collecting material by non-lethal methods. Scientific collections (e.g., preserved plant, animal, and microbial specimens, living organisms, frozen tissues and DNA, living cell lines) have a pivotal potential role in understanding pathogens origins, distribution, and identification of reservoirs [20]. After the first signs of a disease outbreak, public health officials need to determine these factors in a race against time, and scientific collections can offer this information. Although there are pending efforts to standardize sample collection protocols, vouchering of host material, pathogen preparations, or metadata that accompanies such collected materials, the collection of whole specimens is crucial for the study of emerging infectious diseases [21].

Mitochondrial genes have been used extensively to study the evolutionary relationships among bats. Even in the genomic era, the use of single mitochondrial loci is still employed for the rapid identification of species or species complexes. The cytochrome *b* (Cytb) gene is a

widely available marker that produced species-level phylogenies in several mammal groups, including Chiroptera [22–25]. This marker has been used as well in numerous studies involving particular groups of bats [26–28] and was recently implemented in protocols for rapid molecular species identification of bats in eco-epidemiological surveillance [29].

This study aims to discuss the extent of the use of the mitochondrial cytochrome *b* gene for species identification, comparing external morphology identification with a molecular phylogenetic classification based on this marker, and current bat systematics. We used as a case study a set of samples collected in a Rabies virus eco-epidemiological survey involving five localities in the province of Santa Fe (Argentina), where 24 species from 3 families (Phyllostomidae, Molossidae, and Vespertilionidae) and 12 genera have been reported [14, 15, 30, 31]. We further obtained sequences from museum vouchers to test the accuracy of Cytb-based phylogenies in taxonomic identification of bats occurring in the Pampean/Chacoan regions of Argentina. We contrast our results with current systematics of all studied genera based on molecular, morphological, integrative, or genomic approaches showing that Cytb is useful for species-level delimitation in non-conflicting genera (*Eumops*, *Dasypterus*, *Molossops*) and has infrageneric resolution in taxonomically challenging lineages (*Eptesicus*, *Myotis*, *Molossus*). We discuss four sources of incongruence that may act separately or in combination: 1) molecular processes, 2) biology, 3) limitations in identification, and 4) errors in the current taxonomy. The present study confirms the general applicability of cytochrome *b* -based phylogenies in eco-epidemiological studies, but its resolution and reliability depend mainly, but not solely, on the level of genetic differentiation of each bat genus.

## Materials and methods

### Sampling and external morphology identification

The first group of samples corresponds to bats captured in the period January-March 2018 in five localities of Santa Fe province, Argentina, in the frame of the project "Ecoepidemiología de patógenos de importancia para la salud pública y animal en fauna sinantrópica del centro de la provincia de Santa Fe" (Res. C..S. N˚ 632/17). Bats were live-captured with six mist nets (9 and 12 m) for 6 hours from sunset in 3-night trapping sessions in each site (Table 1, Fig 1, sites: (1) Santa Fe, (2) Recreo, (3) Esperanza, (5) San José de Rincón, (6) Cululú). Each captured bat was placed in an individual one-use cloth bag before handling and subsequently subjected to oropharyngeal swabbing for rabies diagnosis. Initial taxonomic identification was done according to external and cranial measurements following Barquez and Díaz [15] and Díaz et al. [16]. For DNA extraction, wing membrane tissue samples were obtained using a biopsy punch (3 mm) and then preserved in 96% ethanol. Sex, reproductive condition [15], and relative age [32, 33] were recorded for each animal. A rehydration solution (dextrose 10%) was orally administered to each bat, after which was marked by a haircut on the back to record recaptures, and finally released [34]. A subgroup of six bats was euthanized and submitted to the Museo Provincial de Ciencias Naturales "Florentino Ameghino" (MFA) (Santa Fe, Argentina) (Table 1). Vouchers were prepared to preserve skin, skull, and skeleton, and ethanol-preserved tissue samples. Handling and preparation methods were approved by the Ethics and Safety Committee of the Universidad Nacional del Litoral (Exp. FCV-0869428-17). The license for collecting samples and specimens was provided by the Ministerio de Medio Ambiente de la Provincia de Santa Fe (Resol. N˚ 093/2018).

The second group of ethanol-preserved tissue samples was obtained from museum vouchers used as reference specimens. Patagium or muscle samples of a total of 25 vouchers from the Santa Fe province (and two from the neighboring Entre Ríos province), deposited at collections in Museo Provincial de Ciencias Naturales "Florentino Ameghino" (MFA) (Santa Fe,

**Table 1. Molecular and morphological identification of bats captured in five localities from Santa Fe.**

| ID (mr) | Department | Locality | Site | Latitude | Longitude | Voucher (MFA) | Sequence length (bp) | External morphology | Phylogeny |
|---|---|---|---|---|---|---|---|---|---|
| 118 | Las Colonias | Esperanza | Sociedad Rural 'Las Colonias' | 31° 25' 32.37" S | 60° 59' 28.31" W | MFA-ZV-M:1408 | 1114 | *Eumops glaucinus* | *Eumops glaucinus* |
| 146 | La Capital | San José de Rincón | Villa California, campo V. Lastra | 31° 36' 07.59" S | 60° 36' 00.25" W | | 957 | *Eumops bonariensis* | *Eumops patagonicus* |
| 151 | La Capital | San José de Rincón | Villa California, campo V. Lastra | 31° 36' 07.59" S | 60° 36' 00.25" W | | 1108 | *Eumops bonariensis* | *Eumops patagonicus* |
| 153 | La Capital | San José de Rincón | Villa California, campo V. Lastra | 31° 36' 07.59" S | 60° 36' 00.25" W | | 1108 | *Eumops bonariensis* | *Eumops bonariensis* |
| 201 | La Capital | Recreo | Train railway | 31° 30' 00.00" S | 60° 43' 59.87" W | | 1113 | *Eumops perotis* | *Eumops perotis* |
| 134 | Las Colonias | Cululú | Arroyo Cululú | 31° 21' 41.94" S | 60° 56' 57.44" W | | 1140 | *Eptesicus diminutus* | *Eptesicus spp.* [1] |
| 140 | Las Colonias | Cululú | Arroyo Cululú | 31° 21' 41.94" S | 60° 56' 57.44" W | | 1103 | *Eptesicus diminutus* | *Eptesicus spp.* [1] |
| 142 | Las Colonias | Cululú | Arroyo Cululú | 31° 21' 41.94" S | 60° 56' 57.44" W | MFA-ZV-M:1492 | 1140 | *Eptesicus furinalis* | *Eptesicus spp.* [1] |
| 195 | Santa Fe | Santa Fe | Sociedad Rural de Santa Fe | 31° 37' 57.75" S | 60° 42' 44.31" W | MFA-ZV-M:1491 | 859 | *Eptesicus furinalis* | *Eptesicus spp.* [1] |
| 124 | Las Colonias | Esperanza | Ciudad | 31° 25' 32.37" S | 60° 59' 28.31" W | MFA-ZV-M:1490 | 1109 | *Dasypterus ega* | *Dasypterus ega* |
| 125 | Las Colonias | Esperanza | Sociedad Rural 'Las Colonias' | 31° 25' 32.37" S | 60° 59' 28.31" W | | 1116 | *Molossops temminckii* | *Molossops temminckii* |
| 202 | La Capital | Recreo | Train railway | 31° 30' 00.00" S | 60° 43' 59.87" W | | 1114 | *Molossops temminckii* | *Molossops temminckii* |
| 145 | Las Colonias | Cululú | Arroyo Cululú | 31° 21' 41.94" S | 60° 56' 57.44" W | | 1114 | *Molossops temminckii* | *Molossops temminckii* |
| 197 | La Capital | Santa Fe | Sociedad Rural de Santa Fe | 31° 37' 57.75" S | 60° 42' 44.31" W | | 1114 | *Promops nasutus* | *Molossus molossus* [2] |
| 129 | Las Colonias | Esperanza | Sociedad Rural 'Las Colonias' | 31° 25' 32.37" S | 60° 59' 28.31" W | MFA-ZV-M:1494 | 1118 | *Molossus molossus* | *Molossus molossus* [2] |
| 130 | Las Colonias | Esperanza | Sociedad Rural 'Las Colonias' | 31° 25' 32.37" S | 60° 59' 28.31" W | | 1108 | *Molossus molossus* | *Molossus molossus* [2] |
| 177 | La Capital | Santa Fe | Puente Parque Sur | 31° 39' 31.22" S | 60° 42' 24.78" W | | 1114 | *Molossus molossus* | *Molossus molossus* [2] |
| 190 | La Capital | Santa Fe | Puente Parque Sur | 31° 39' 31.22" S | 60° 42' 24.78" W | | 1116 | *Molossus molossus* | *Molossus molossus* [2] |
| 192 | La Capital | Santa Fe | Sociedad Rural de Santa Fe | 31° 37' 57.75" S | 60° 42' 44.31" W | | 1111 | *Molossus molossus* | *Molossus molossus* [2] |
| 193 | La Capital | Santa Fe | Sociedad Rural de Santa Fe | 31° 37' 57.75" S | 60° 42' 44.31" W | | 1109 | *Molossus molossus* | *Molossus molossus* [2] |
| 194 | La Capital | Santa Fe | Sociedad Rural de Santa Fe | 31° 37' 57.75" S | 60° 42' 44.31" W | | 1113 | *Molossus molossus* | *Molossus molossus* [2] |
| 196 | La Capital | Santa Fe | Sociedad Rural de Santa Fe | 31° 37' 57.75" S | 60° 42' 44.31" W | | 1120 | *Molossus molossus* | *Molossus molossus* [2] |
| 198 | La Capital | Santa Fe | Sociedad Rural de Santa Fe | 31° 37' 57.75" S | 60° 42' 44.31" W | | 1111 | *Molossus molossus* | *Molossus molossus* [2] |
| 199 | La Capital | Santa Fe | Sociedad Rural de Santa Fe | 31° 37' 57.75" S | 60° 42' 44.31" W | | 1113 | *Molossus molossus* | *Molossus molossus* [2] |
| 123 | Las Colonias | Esperanza | Sociedad Rural 'Las Colonias' | 31° 25' 32.37" S | 60° 59' 28.31" W | MEM 236* | 1112 | *Myotis spp.* | *Myotis nigricans* [3] |
| 126 | Las Colonias | Esperanza | Sociedad Rural 'Las Colonias' | 31° 25' 32.37" S | 60° 59' 28.31" W | | 1110 | *Myotis spp.* | *Myotis nigricans* [3] |

*(Continued)*

**Table 1.** (Continued)

| ID (mr) | Department | Locality | Site | Latitude | Longitude | Voucher (MFA) | Sequence length (bp) | External morphology | Phylogeny |
|---|---|---|---|---|---|---|---|---|---|
| 133 | Las Colonias | Esperanza | Sociedad Rural 'Las Colonias' | 31° 25' 32.37" S | 60° 59' 28.31" W | | 1108 | *Myotis spp.* | *Myotis nigricans* [3] |
| 144 | Las Colonias | Cululú | Arroyo Cululú | 31° 21' 41.94" S | 60° 56' 57.44" W | | 1120 | *Myotis levis* | *Myotis nigricans* [3] |
| 148 | La Capital | San José de Rincón | Villa California, campo V. Lastra | 31° 36' 07.59" S | 60° 36' 00.25" W | | 1110 | *Myotis spp.* | *Myotis nigricans* [3] |
| 154 | La Capital | San José de Rincón | Villa California, campo V. Lastra | 31° 36' 07.59" S | 60° 36' 00.25" W | | 1122 | *Myotis cf nigricans* | *Myotis nigricans* [3] |
| 155 | La Capital | San José de Rincón | Villa California, campo V. Lastra | 31° 36' 07.59" S | 60° 36' 00.25" W | | 1114 | *Myotis cf nigricans* | *Myotis nigricans* [3] |
| 156 | La Capital | San José de Rincón | Villa California, campo V. Lastra | 31° 36' 07.59" S | 60° 36' 00.25" W | | 1115 | *Myotis cf nigricans* | *Myotis nigricans* [3] |
| 159 | La Capital | Santa Fe | Destacamento de Vigilancia Cuartel 'Guadalupe' | 31° 35' 28.53" S | 60° 40' 25.33" W | | 1107 | *Myotis spp.* | *Myotis nigricans* [3] |

[1]: *E. diminutus* and *E. furinalis* are not reciprocally monophyletic

[2]: *M. molossus*, *M. rufus*, *M. currentium*, and *M. bondae* are not reciprocally monophyletic

[3]: *M. nigricans*, *M. levis*, *M albescens*, *M. ruber*, and *M. riparius* are not reciprocally monophyletic

[*] Collector's personal ID

NA: no amplification

Identification numbers (ID), geolocalization data, and sequence length are shown.

Argentina) and Museo Provincial de Ciencias Naturales "Dr. Ángel Gallardo" (MG) (Rosario, Argentina) were included. These vouchers were revised, identified, and/or re-identified based on Barquez et al. [35], Barquez and Díaz [15], and Díaz et al. [16]. All the specimens of the Museo Ameghino were provided by the Dirección General de Bioquímica y Farmacia, Laboratorio Central, Santa Fe province, framed in the RAVB passive surveillance activities of the institution, for which no capture site or georeference is provided (Table 2).

## DNA extraction, primer design, and PCR amplification

Total genomic DNA was extracted from wing membrane or muscle samples using the High Pure PCR Template Preparation Kit (Roche Life Science®). The complete mitochondrial cytochrome *b* (Cytb) gene was amplified by PCR and sequenced using primers Bat 05A (sense: 5′–CGACTAATGACATGAAAAATCACCGTTG–3′, Tm: 63.2°C) and Bat 14A (antisense: 5′–TATTCCCTTTGCCGGTTTACAAGACC–3′, Tm: 64.6°C), designed based on the complete mitochondrial genome of the phyllostomid *Artibeus jamaicensis* [36]. This primer pair yielded PCR products in all assayed genera except for *Eptesicus*, as revealed by agarose gel electrophoresis.

We further inspected primer sequence mismatches in different bat mitogenomes. To this end, we performed a discontiguous megablast search using a portion of the mitogenome of *A. jamaicensis* as query. Since the complete Cytb sequence is between positions 14150 and 15289 of the *A. jamaicensis* mitochondrial DNA (mtDNA) genome, we blasted in the range 13700–15500 and kept 19 sequences corresponding to genera occurring in America (*Eptesicus*, *Lasiurus*, *Myotis*, *Tadarida*, *Artibeus* and *Diaemus*). We verified that the region targeting primer Bat 05A was more conserved in all retrieved sequences (mismatch mean = 2.27, median = 2), compared with Bat 14A (mismatch mean = 5.05, median = 5). Additionally, there

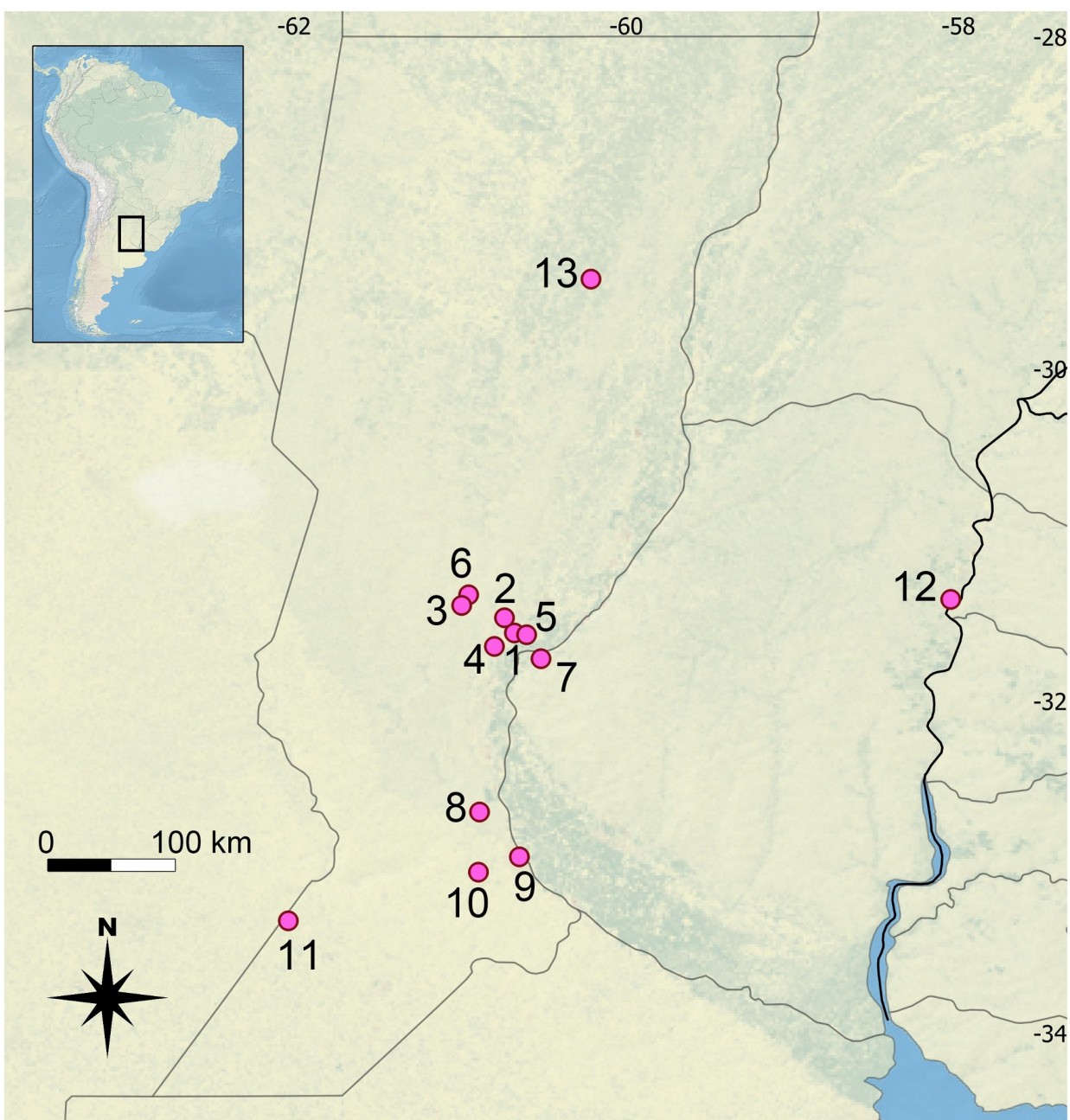

**Fig 1. Sampling localities.** Map showing sampling localities: Santa Fe (1), Recreo (2), Esperanza (3), Santo Tomé (4), San José de Rincón (5), Cululú (6), Paraná (7), Pueblo Andino (8), Rosario (9), Zavalla (10), Chañar caído (11), Concordia (12), and Vera (13).

were gaps in the region corresponding to Bat 14A in three sequences from *Artibeus* (2) and *Myotis* (1). Hence, we designed an alternative primer, to replace Bat 14A using a conserved region among the 19 mitogenomes. This primer was named Bat-Ep (antisense: 5' -TAGTTT AABTAGAAYHYCAGCTTTGGG-3', Tm: 61.6) (S1 File).

Amplification of the Cytb gene from 5 μL of DNA was carried out in a final volume of 50 μL using either Bat 05A/Bat 14A or Bat 05A/Bat-Ep primers, 10 X PCR buffer, 1.5 mM MgCl₂, 200 μM of each dNTP, 2 μM of each primer and 0.4 units of Taq DNA polymerase.

**Table 2. Molecular and morphological identification of voucher specimens.**

| Voucher | Sequence length (bp) | Species | Phylogeny | Province | Department | Locality | Site | Latitude | Longitude |
|---|---|---|---|---|---|---|---|---|---|
| MFA-ZV-M: 1360 | 1112 | *Eumops patagonicus* | *Eumops patagonicus* | Santa Fe | La Capital | Santo Tomé* | | 31° 40' 20.53" S | 60° 47' 38.68" W |
| MFA-ZV-M: 1411 | 1140 | *Eumops patagonicus* | *Eumops patagonicus* | Santa Fe | La Capital | Santa Fe* | | 31° 37' 10.50" S | 60° 42' 07.70" W |
| MFA-ZV-M: 1415 | 1117 | *Eumops patagonicus* | *Eumops patagonicus* | Santa Fe | Rosario | Rosario* | | 32° 57' 32.77" S | 60° 43' 19.37" W |
| MFA-ZV-M: 1449 | 1124 | *Eumops perotis* | *Eumops perotis* | Santa Fe | La Capital | Santa Fe* | | 31° 37' 10.50" S | 60° 42' 07.70" W |
| MG-ZV-M: 199 | 1118 | *Eumops perotis* | *Eumops perotis* | Santa Fe | Rosario | Zavalla | Parque Villarino, Facultad de Ciencias Agrarias, UNR | 33° 01' 54.19" S | 60° 53' 24.90" W |
| MG-ZV-M: 219 | 1108 | *Eumops patagonicus* | *Eumops patagonicus* | Santa Fe | Rosario | Zavalla | Parque Villarino, Facultad de Ciencias Agrarias, UNR | 33° 01' 33.12" S | 60° 53' 10.89" W |
| MG-ZV-M: 224 | 1112 | *Eumops patagonicus* | *Eumops patagonicus* | Santa Fe | Rosario | Zavalla | Parque Villarino, Facultad de Ciencias Agrarias, UNR | 33° 01' 33.12" S | 60° 53' 10.89" W |
| MFA-ZV-M: 1461 | 901 | *Eptesicus diminutus* | *Eptesicus spp*[1] | Santa Fe | Las Colonias | Esperanza* | | 31° 27' 03.16" S | 60° 55' 44.29" W |
| MFA-ZV-M: 1421 | 927 | *Eptesicus furinalis* | *Eptesicus spp*[1] | Santa Fe | Las Colonias | Esperanza* | | 31° 27' 03.16" S | 60° 55' 44.29" W |
| MG-ZV-M: 175 | 933 | *Eptesicus furinalis* | *Eptesicus spp*[1] | Santa Fe | Rosario | Rosario | Tucumán 4176 | 32° 56' 02.96" S | 60° 40' 44.64" W |
| MG-ZV-M: 184 | 928 | *Eptesicus furinalis* | *Eptesicus spp*[1] | Santa Fe | Rosario | Rosario | Tucumán 4176 | 32° 56' 02.96" S | 60° 40' 44.64" W |
| MG-ZV-M: 164 | 424 | *Dasypterus ega* | *Dasypterus ega* | Santa Fe | Iriondo | Pueblo Andino | Centro Comunitario | 32° 40' 13.37" S | 60° 52' 28.56" W |
| MG-ZV-M: 166 | 1009 | *Dasypterus ega* | *Dasypterus ega* | Santa Fe | Iriondo | Pueblo Andino | Cancha fútbol | 32° 40' 23.94" S | 60° 52' 28.44" W |
| MFA-ZV-M: 1371 | 1108 | *Molossops temminckii* | *Molossops temminckii* | Entre Ríos | Concordia | Concordia* | | 31° 23' 21.59" S | 58° 02' 45.54" W |
| MFA-ZV-M: 1413 | 970 | *Molossus molossus* | *Molossus molossus*[2] | Entre Ríos | Paraná | Paraná* | | 31° 44' 47.12" S | 60° 30' 48.93" W |
| MFA-ZV-M: 1414 | 953 | *Molossus molossus* | *Molossus molossus*[2] | Santa Fe | Rosario | Rosario* | | 32° 58' 05.95" S | 60° 40' 54.76" W |
| MFA-ZV-M: 1423 | 995 | *Molossus rufus* | *Molossus rufus*[2] | Santa Fe | Vera | Vera* | | 29° 27' 38.90" S | 60° 12' 44.78" W |
| MFA-ZV-M: 1431 | 1059 | *Molossus molossus* | *Molossus molossus*[2] | Santa Fe | La Capital | Santa Fe* | | 31° 37' 10.50" S | 60° 42' 07.70" W |
| MFA-ZV-M: 1435 | 1114 | *Molossus molossus* | *Molossus molossus*[2] | Santa Fe | La Capital | Santa Fe* | | 31° 37' 10.50" S | 60° 42' 07.70" W |
| MG-ZV-M: 176 | 1112 | *Molossus molossus* | *Molossus molossus*[2] | Santa Fe | Iriondo | Pueblo Andino | Reserva Hídrica 'Río Carcarañá' | 32° 40' 10.56" S | 60° 53' 01.07" W |
| MG-ZV-M: 208 | 900 | *Molossus molossus* | *Molossus molossus*[2] | Santa Fe | Rosario | Zavalla | Parque Villarino, Facultad de Ciencias Agrarias, UNR | 33° 01' 47.20" S | 60° 53' 19.70" W |
| MG-ZV-M: 225 | 753 | *Molossus molossus* | *Molossus molossus*[2] | Santa Fe | Rosario | Zavalla | Parque Villarino, Facultad de Ciencias Agrarias, UNR | 33° 01' 47.20" S | 60° 53' 19.70" W |
| MG-ZV-M: 296 | 1113 | *Molossus molossus* | *Molossus molossus*[2] | Santa Fe | Rosario | Rosario | Presidente Roca 300 | 32° 56' 21.82" S | 60° 38' 37.20" W |
| MFA-ZV-M: 1425 | 1115 | *Myotis nigricans* | *Myotis nigricans*[3] | Santa Fe | Las Colonias | Esperanza* | | 31° 27' 03.16" S | 60° 55' 44.29" W |
| MG-ZV-M: 217 | 1111 | *Myotis albescens* | *Myotis levis*[3] | Santa Fe | Rosario | Zavalla | Parque Villarino, Facultad de Ciencias Agrarias, UNR | 33° 01' 33.12" S | 60° 53' 10.89" W |
| MG-ZV-M: 221 | 1134 | *Myotis levis* | *Myotis levis*[3] | Santa Fe | Rosario | Zavalla | Parque Villarino, Facultad de Ciencias Agrarias, UNR | 33° 01' 53.76" S | 60° 53' 34.02" W |

*(Continued)*

**Table 2.** (Continued)

| Voucher | Sequence length (bp) | Species | Phylogeny | Province | Department | Locality | Site | Latitude | Longitude |
|---|---|---|---|---|---|---|---|---|---|
| MG-ZV-M: 233 | 1118 | *Myotis levis* | *Myotis levis*[3] | Santa Fe | Caseros | Chañar Ladeado | San Martín 1060 | 33˚ 19' 28.55" S | 62˚ 02' 06.35" W |

Molecular and morphological identification of voucher specimens deposited at the collections of the Museo Provincial de Ciencias Naturales "Florentino Ameghino", Santa Fe (MFA-ZV-M) and Museo Provincial de Ciencias Naturales "Dr. Ángel Gallardo", Rosario (MG-ZV-M). Identification numbers (ID), geolocalization data, and sequence length are shown.

MFA-ZV-M (Museo Provincial de Ciencias Naturales "Florentino Ameghino", Santa Fe)

MG-ZV-M (Museo Provincial de Ciencias Naturales "Dr. Ángel Gallardo", Rosario)

[1]: *E. diminutus* and *E. furinalis* are not reciprocally monophyletic

[2]: *M. molossus*, *M. rufus*, *M. currentium*, and *M. bondae* are not reciprocally monophyletic

[3]: *M. nigricans*, *M. levis*, *M albescens*, *M. ruber*, and *M. riparius* are not reciprocally monophyletic

[*]: Approximate georeference

Cycling conditions were 1 cycle of 5 min at 94˚C, 35 cycles at 94˚C for 45 s, 55˚C for 45 s, and 72˚C for 2 min followed by a final extension step at 72˚C for 10 min. After each run, PCR products were electrophoresed on 1% agarose gel in TBE buffer (1x 0.1 M Tris, 0.09 M boric acid, and 0.001 M EDTA) containing 0.5 mg/L ethidium bromide and visualized on a transilluminator under UV light. Total PCR products were purified using the High Pure PCR Product Purification Kit (Roche Life Science®).

## Sequencing, sequence retrieval, and alignment

DNA sequencing was performed using 4 μL of BigDye 3.1 (Applied Biosystems Inc. Foster City, California, USA), 4 μL of 5x sequencing buffer (Applied Biosystems Inc. Foster City, California, USA), 3.2 pmol of each primer (sense and antisense, as described in section 2.2), 30–50 ng of target DNA and DNAse/RNAse-free water to a final reaction volume of 10 μL. Reactions were carried out in a Proflex Thermal Cycler (Applied Biosystems Inc. Foster City, California, USA) under the following cycling conditions: 35 cycles at 96˚C for 10 s, 50˚C for 5 s, and 60˚C for 4 min. After the sequencing reaction, samples were precipitated and dried using 100% ethanol, followed by 70% ethanol, and centrifuging. After precipitation, sequences were generated on an ABI PRISM 310 Genetic Analyzer (Applied Biosystems Inc. Foster City, California, USA). The sequences generated in this study were deposited in Genbank under Accession Numbers MT262814—MT262873.

In addition to the samples obtained in the field, and those from museums, we included a total of 142 sequences retrieved from GenBank (S2 File), to enrich the analysis and test species monophyly. We included members of all genera of the species that have verified distributions in the capture zone, including also allied species aiming to achieve the broadest possible representation of their geographical range, with special emphasis on South American species, but also including vouchers from Central and North America. Whenever possible, we took the precaution to include only those sequences that have a counterpart at a museum collection; that is, only those sequences from specimens submitted as vouchers, susceptible to being reanalyzed morphologically or molecularly. Additionally, three candidate outgroups were included: *Furipterus* (1) and *Macrotus* (2), the last one representing non-Argentinian outgroups for Phyllostomidae, a family which also is an outgroup in our dataset since we captured members of Vespertilionidae and Molossidae solely, while the former representing Furipteridae, an outgroup for all families occurring in Argentina [25].

The final matrix contained 202 Cytb nucleotide sequences representing 41 species and 14 genera, which were aligned using Clustal Omega [37]. No deletions, insertions or stop codons were observed in the 1140 bp alignment (S3 File).

## Phylogenetic analysis

The phylogenetic analysis was performed in MrBayes 3.2.6 [38] on the CIPRES Science Gateway [39], partitioning the Cytb sequence in 1st+2nd and 3rd codon positions separately. Nucleotide substitution models were estimated using JModeltest2 [40] under the Akaike Information Criterion (correcting by the number of taxa). For the partition comprising 1st and 2nd codon positions, the selected substitution model was HKY+I+G, while HKY was selected for the 3rd codon position. Two independent runs for $2x10^8$ MCMC (Markov chain Monte Carlo) generations, sampling every $2x10^4$ generations were carried out. The sequence corresponding to *Furipterus horrens* was set as outgroup (GenBank Accession number: AY621004).

Convergence was assessed by analyzing the potential scale reduction factor (PSRF), and the average standard deviation of split frequencies (ASDSF). The *burnin* phase was set up in the generation that fulfilled PSRF values of 1.00–1.02 for all estimated parameters and standard deviations lower than 0.01, which corresponded to 5.75% of the total run. Trees were visualized with iTOL [41] and Figtree [42].

## Results

### Identification based on external morphology

A total of 33 captured adult bats belonging to 11 species, seven genera, and two families (Vespertilionidae and Molossidae) (Table 1) were identified based on morphological traits following Barquez et al. [35], Barquez and Díaz [15], and Díaz et al. [16]. One specimen could not be identified using the keys beyond the genus level (*Myotis* spp.) (Table 1). Regarding museum vouchers (Table 2), 11 species from six genera were identified following the identification keys mentioned above.

### Cytb amplification with primers Bat 05A and Bat-Ep

The combination of primers Bat 05A and Bat-Ep successfully amplified the Cytb gene and flanking regions, yielding a PCR product of ca 1330 bp (S1 Fig). We used these primers to amplify the Cytb sequence of *Eptesicus* but also screened its applicability to other genera such as *Dasypterus*, *Molossus*, *Molossops*, *Myotis*, and *Eumops*. In some samples, a minor product of ca 500 bp was also amplified, however, it was lost after the PCR-product purification procedure and hence did not interfere in sequencing. This was not the case for the sample corresponding to *Eumops dabbenei*, in which an extra band of ca 700 bp, and several minor bands distorted the sequence read, and could not be recovered after band excision and purification. In total, 13 sequences were obtained with the primers set Bat 05A and Bat-Ep (Tables 1 and 2). The remaining 47 sequences were obtained with primers Bat 05A and Bat 14A (Tables 1 and 2).

### Phylogenetic analysis

The complete phylogenetic tree is shown in Fig 2 (the interactive version is hosted on the iTOL website: https://itol.embl.de/tree/181461382415312158507582). The relationships between and within families Phyllostomidae, Molossidae, and Vespertilionidae were congruent with current systematics [25]. At the genus level, there are cases in which species achieve reciprocal monophyly, but also many exceptions.

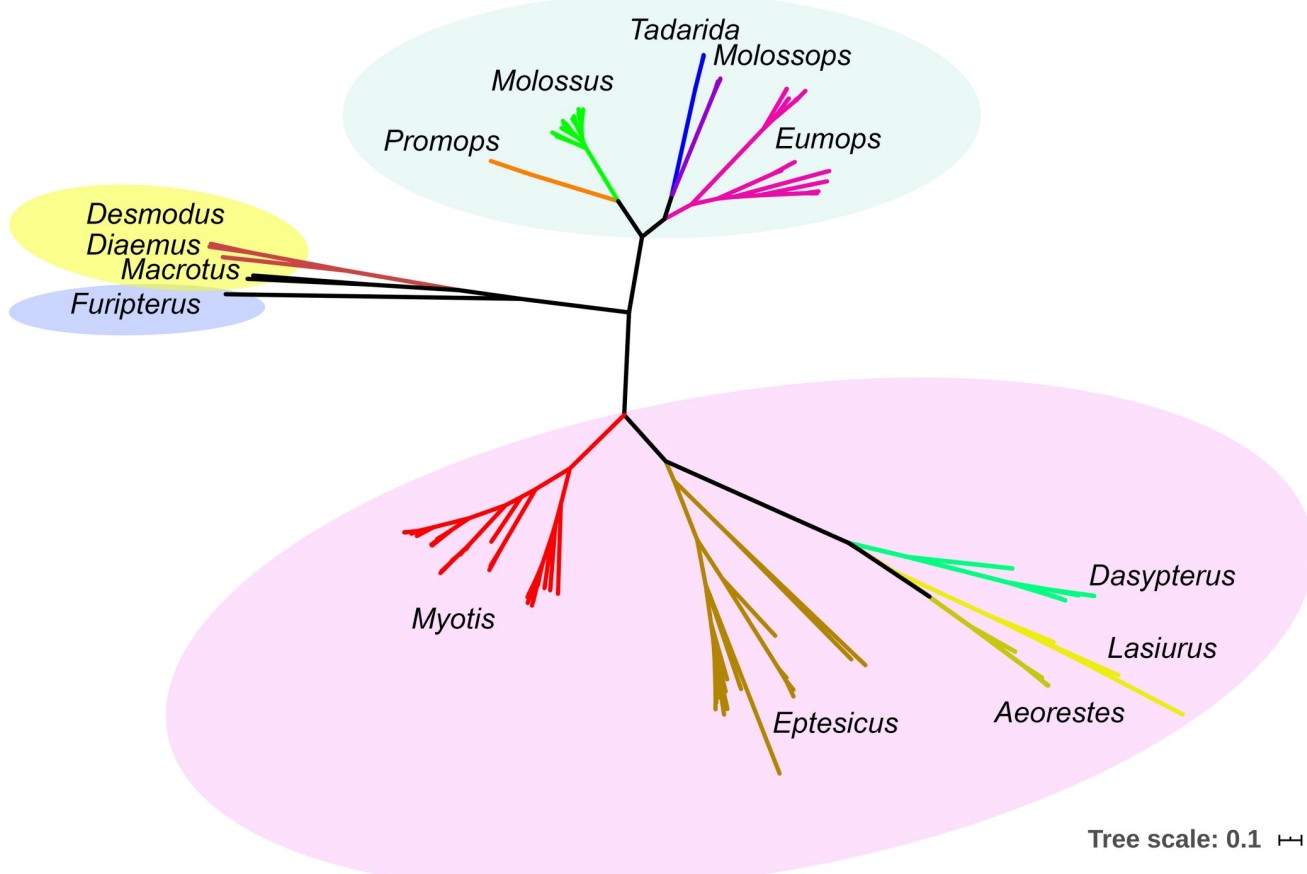

**Fig 2. Cytochrome *b*-based Bayesian phylogeny of bats at the genus/family level.** Families are depicted with colored ovals: Phyllostomidae (yellow), Molossidae (green), Vespertilionidae (purple), and Furipteridae (blue). The scale units are substitutions per site. All nodes supporting genus-level lineages have Bayesian Posterior Probability = 0.1.

Within *Eumops*, all sister species showed reciprocal monophyly (Fig 3). Field samples morphologically identified as *Eumops perotis* and *Eumops glaucinus* were confirmed by the phylogeny, as occurred with museum vouchers identified as *E. perotis* and *Eumops patagonicus*. Two of the three captured specimens initially identified as *Eumops bonariensis*, fell into the *E. patagonicus* clade, while the remnant fell within *E. bonariensis*.

The genus *Dasypterus* was monophyletic (Fig 4) and sister to the clade (*Aeorestes*, *Lasiurus*). Both field specimens and museum vouchers assigned to *Dasypterus ega* pertain to an exclusive monophyletic group (Fig 4). The three specimens captured in the field and the museum voucher identified as *Molossops temminckii* fell into a monospecific clade (Fig 5).

The four field specimens and the four vouchers identified as *Eptesicus furinalis* or *Eptesicus diminutus* were intermixed in the phylogeny (Fig 6). Taken together, these sequences are monophyletic and belong to a broader group that includes exclusively *E. furinalis* and *E. diminutus* GenBank sequences.

Within *Myotis*, except for *Myotis ruber*, none of the species was monophyletic (Fig 7). Moreover, the species *Myotis riparius*, *Myotis nigricans*, *Myotis albescens*, and *Myotis levis*, were polyphyletic, although field specimens and museum vouchers fell in a clade that excluded *M. riparius*. Notably, all specimens captured at the field fell within an exclusive *M. nigricans* clade and depicted extremely low variability as revealed by its short branch lengths.

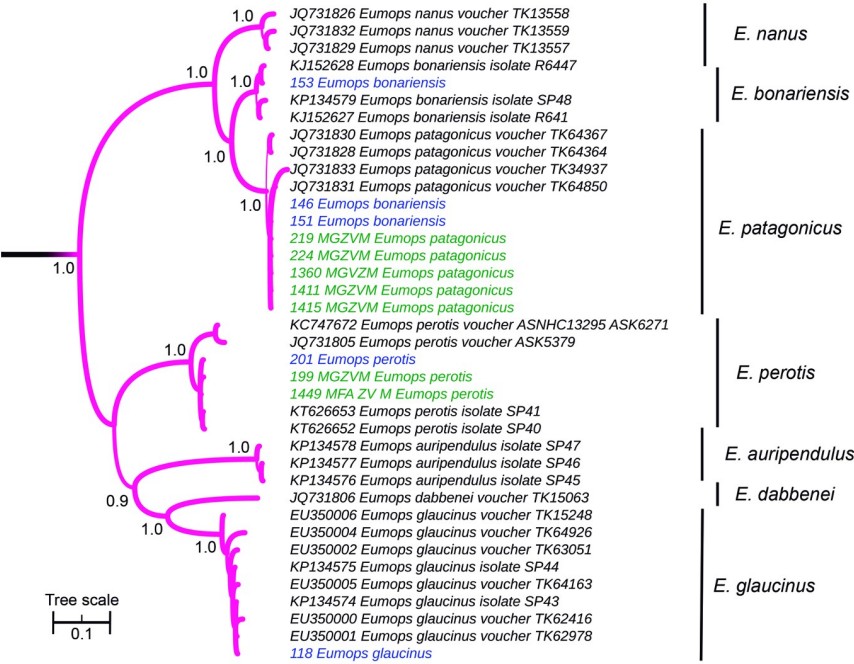

**Fig 3. Cytochrome *b*-based Bayesian phylogeny of *Eumops*.** Main clades posterior probability values are annotated. Branch width represents internal node support. Terminal font color indicates sequence source: black indicates sequences retrieved from Genbank, green indicates museum voucher sequences obtained in this study, while blue color indicates samples captured at the field and sequenced in this study. The scale units are substitutions per site.

Within *Molossus*, the three species, *Molossus molossus*, *Molossus currentium*, and *Molossus rufus*, were polyphyletic (Fig 8). The whole genus split into four main clades, two of which included exclusively *M. molossus*. Field samples identified as *M. molossus* fell in three of the four main clades of the genus. The same occurred with museum vouchers ascribed to *M. molossus*. The voucher of *M. rufus* pertains to a clade that is almost exclusively composed of *M. rufus* and *M. currentium*. Field sample 197 was morphologically identified as *Promops* but fell

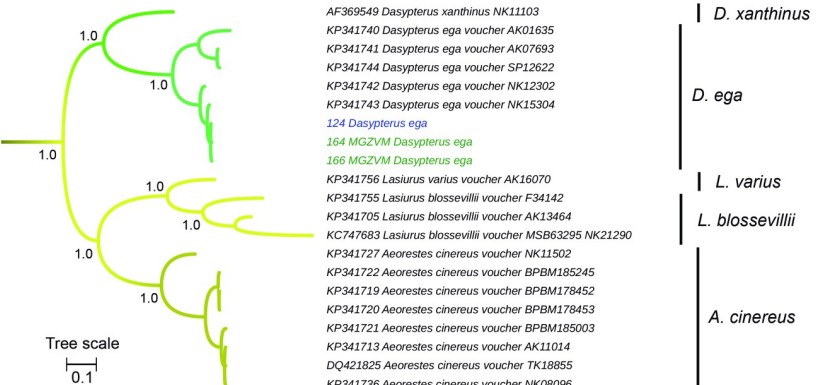

**Fig 4. Cytochrome *b*-based Bayesian phylogeny of *Dasypterus*, *Lasiurus*, and *Aeorestes*.** Branch width represents internal node support. Terminal font color indicates sequence source: black indicates sequences retrieved from Genbank, green indicates museum voucher sequences obtained in this study, while blue color indicates samples captured at the field and sequenced in this study. The scale units are substitutions per site. Main clades posterior probability values are annotated.

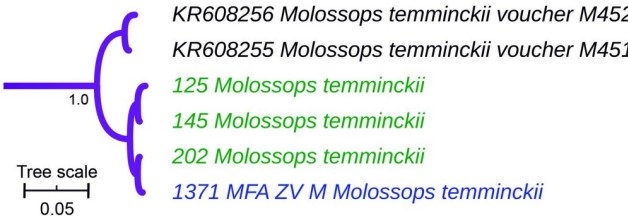

**Fig 5. Cytochrome *b*-based Bayesian phylogeny of *Molossops*.** Branch width represents internal node support. Terminal font color indicates sequence source: black indicates sequences retrieved from Genbank, green indicates museum voucher sequences obtained in this study, while blue color indicates samples captured at the field and sequenced in this study. The scale units are substitutions per site. Main clades posterior probability values are annotated.

within one of the *M. molossus* exclusive clades. All *Molossus* sequences depict short branch lengths, denoting low levels of genetic differentiation within this genus.

## Discussion

Incongruence between morphological and molecular identification may originate from methodological problems or biological problems, depending on the causes of that conflict. Four sources of discrepancy may act separately or in combination: 1) molecular processes, 2) biology, 3) limitations in morphological identification, and 4) errors in the current taxonomy. We will examine each of these causes of conflict and the feasibility of discerning between them under the applied methodology. We will also discuss the available approaches to overcome, at least partially, these shortcomings. There are different processes at the molecular level (1) that may produce conflict between different data sets, given that genes may not necessarily reflect organismal relationships. Processes such as gene duplication, recombination, and natural selection are examples of molecular causes of gene tree incongruence [43]. However, the most frequent cause of conflict among gene trees is incomplete lineage sorting. The occurrence of this phenomenon is much more widespread, because it does not depend on specific molecular

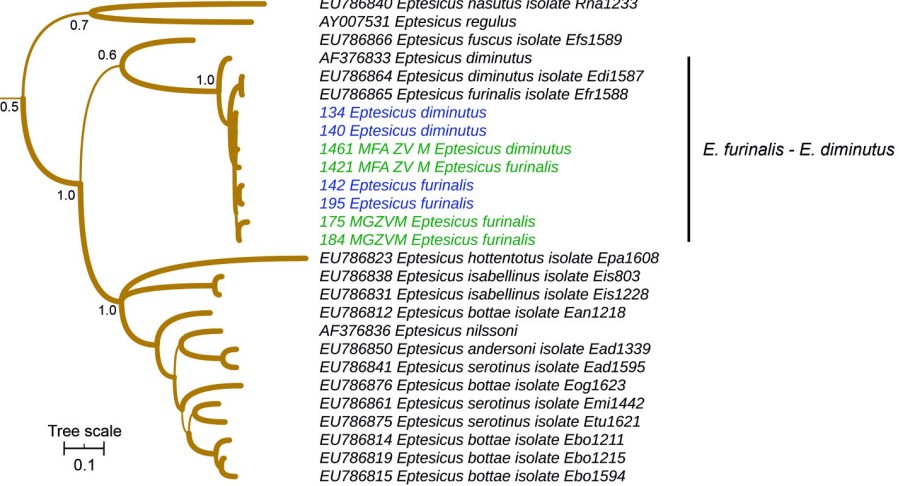

**Fig 6. Cytochrome *b*-based Bayesian phylogeny of *Eptesicus*.** Main clades posterior probability values are annotated. Branch width represents internal node support. Terminal font color indicates sequence source: black indicates sequences retrieved from Genbank, green indicates museum voucher sequences obtained in this study, while blue color indicates samples captured at the field and sequenced in this study. The scale units are substitutions per site.

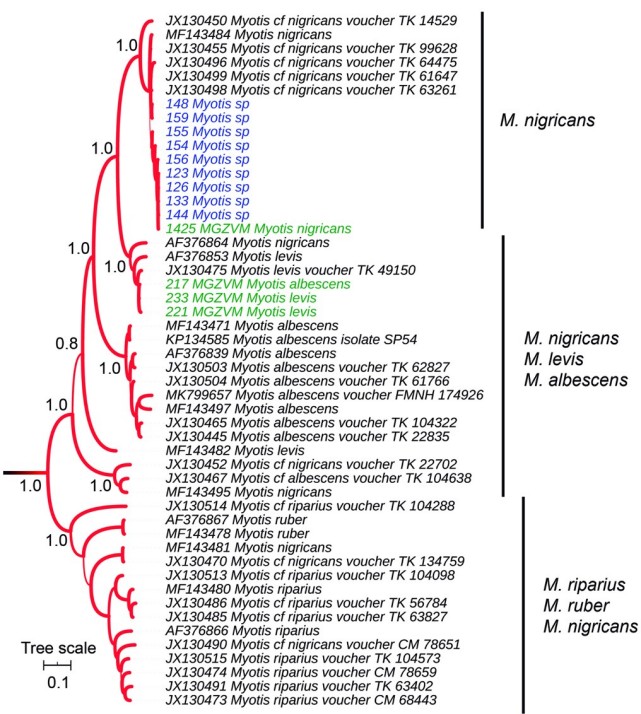

**Fig 7. Cytochrome *b*-based Bayesian phylogeny of *Myotis*.** Branch width represents internal node support. Terminal font color indicates sequence source: black indicates sequences retrieved from Genbank, green indicates museum voucher sequences obtained in this study, while blue color indicates samples captured at the field and sequenced in this study. The scale units are substitutions per site. Main clades posterior probability values are annotated.

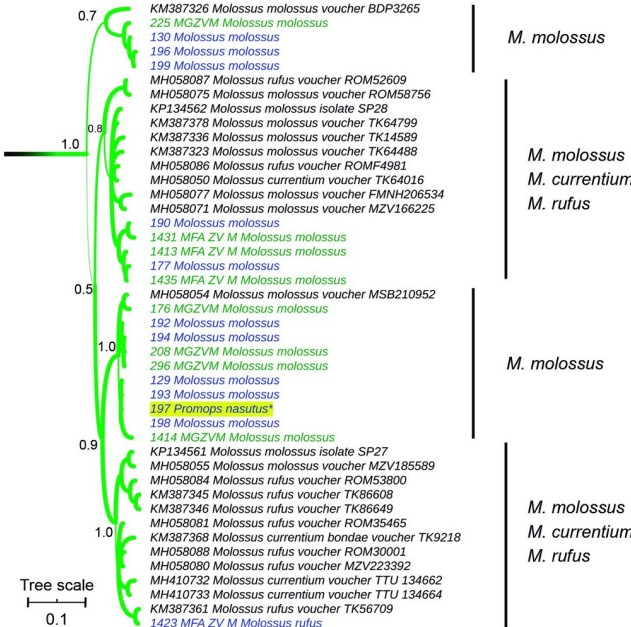

**Fig 8. Cytochrome *b*-based Bayesian phylogeny of *Molossus*.** Main clades posterior probability values are annotated. Branch width represents internal node support. Terminal font color indicates sequence source: black indicates sequences retrieved from Genbank, green indicates museum voucher sequences obtained in this study, while blue color indicates samples captured at the field and sequenced in this study. The scale units are substitutions per site.

events occurring in particular lineages, but instead is determined by intrinsic factors at the population level (i.e. the rate of genetic drift) [44]. This process is particularly common at shallow phylogenetic depths, where the elapsed time between species divergence is too short for ancestral polymorphisms to have sorted into reciprocally monophyletic lineages [45]. Hence, incomplete lineage sorting may produce conflicting phylogenies between independently segregating markers that may also conflict with morphological classification. A highly correlated issue with incomplete lineage sorting is low genetic differentiation, which is also determined by short divergence times. These ubiquitous problems may be overcome with the use of multilocus approaches that apply statistical methods for dealing with gene tree heterogeneity and coalescent stochasticity [44].

Hybridization followed by introgression represents a biological (2) source of conflict in species identification, which may affect both molecular and morphological levels. Introgression can yield species-level non-monophyly by introducing alleles across species boundaries [46]. The rates of mitochondrial and nuclear introgression frequently differ, with some taxa depicting biases for mitochondrial and others for nuclear genes. The bias towards mitochondrial introgression is more frequent in XY sex-determination systems (bats) [47].

The distinction between incomplete lineage sorting and introgression is a critical task in evolutionary studies. The development of coalescent genealogy samplers has facilitated the estimation of past qualities of a population, such as its size, time of divergence from another population, or immigration rates, in a statistical framework [48, 49]. However, these methods require the use of multiple loci, which exceeds the scope of this study and make these two sources of conflict impossible to discriminate. These limitations do not make single genes (or linked genes such as mtDNA markers) useless, but may not be universally applicable, and their performance should be tested in specific taxonomic groups. Mitochondrial DNA may not be enough for establishing a classification system but may be useful in the context of sample identification, especially in field studies, making it a valuable tool for ecological surveys [50].

Another instance of incongruence is the occurrence of errors in external morphology-based identification (3). As mentioned, morphological methods rely upon taxonomic expertise and are prone to subjective errors, especially in bats where phenotypic plasticity [51] and cryptic taxa are prevalent [7–11]. Among morphological methods, geometric morphometrics has shown greater power for morphologically similar species discrimination [4]. However, although useful for identifying anatomical regions that could serve for guiding which linear measures to take, this approach is not applicable for rapid species identification in the field. A common issue derived from morphological misidentification is the incorrect annotation of a sequence in a database. Given the inherent nature of public databases, it is inevitable that erroneous data will be present, as reported for several taxonomic groups [52–54]. Genbank does not store sequence chromatograms, collection metadata, or photographs of specimens, which may help to identify at least part of annotation errors. To minimize them, we used -when possible- sequences derived from vouchered specimens, identified by taxonomic experts (S2 File).

The last cause of incongruence between genetic and morphological identification is conflicts in current taxonomy (4). Species are essential units of analysis in biology and their delimitation is the most fundamental aspect of systematics. However, there are many distinct, and partially incompatible, epistemological views of the species concept that emerge from considering different biological features [55, 56]. Properties such as reproductive isolation, phenotypic distinction, reciprocal monophyly, or ecological differentiation, to list some of them, are reached after enough time since the divergence of two nascent species. But these biological features evolve at different rates, and not even necessarily in the same order during the process of speciation, and when times of divergence are small, there may be discordant delimitation criteria depending on which species concept is taken into consideration [57]. This is due not only

because there is little differentiation, but also because there is greater incongruence among different character sources [46]. When delimiting species, taxonomists take decisions that could conflict with different lines of evidence. Species taxa supported by several independent and concurring kinds of characters could be considered stable hypotheses. However, species could be–and are frequently- overestimated (taxonomic inflation) [58] or underestimated (taxonomic inertia) [59]. Non-reciprocal monophyly in single-gene phylogenies may not be sufficient for the distinction between taxonomic inflation and incomplete lineage sorting/introgression, but the integration of these results with independent sources of information (multi-locus or genome-based phylogenies) may allow inferring its causes.

In the present study, morphological and molecular identification of field samples, as well as museum vouchers, were congruent at the genus level (Tables 1 and 2, Figs 3–8). The only exception was sample 197, which was identified as *Promops* when captured but was molecularly identified as *Molossus* (Fig 8). During field specimen identification, the action of facial muscles could lead to incorrect classification. In this particular case, the specimen was confused with a member of its sister genus (Fig 2), exemplifying an error in morphological identification (3).

At the infrageneric level, molecular and external morphology-based classification showed varying levels of discrepancy depending on each genus (Tables 1 and 2, Figs 3–8). *Molossops* showed total concordance between both sources of information (Fig 5).

The yellow bats of the genus *Dasypterus* were reciprocally monophyletic with respect to the (*Lasiurus*, *Aeorestes*) clade (Fig 4), in congruence with previously published phylogenies based on mtDNA, nuclear and Y-Chromosome genes [60, 61]. It is worth noting that the sequence of the species *Dasypterus xanthinus* is annotated in Genbank (Accession Number AF369549) as *Lasiurus xanthinus*. The submission of this sequence was made shortly after the first proposal [60] for splitting *Lasiurus* into three genera (*Dasypterus*, *Aeorestes*, *Lasiurus*), constituting an example of inaccuracy in database annotation not because of an error in morphological identification (3), but to an error in taxonomy (4). This occurs also with *Aorestes cinereus* sequences; all of them are annotated as *Lasiurus cinereus* in Genbank, denoting the difficulty in database curation.

Within *Eumops*, all species were reciprocally monophyletic, in concordance with previous Cytb-based phylogenies [62], but there were specimens classified in the field as *E. bonariensis* that were included in the *E. patagonicus* clade according to the Cytb molecular phylogeny (Fig 3). The fact that these two specimens were identified at the field and that, except for them, both species are reciprocally monophyletic [62], raises the possibility that these are indeed *E. patagonicus*, representing another case of morphological misidentification (3). *Eptesicus furinalis* and *E. diminutus*, fell within a clade excluding other *Eptesicus* species (Fig 6), in congruence with previously published phylogenies [63]. The sequences of museum vouchers and field specimens resulted intermingled in the Cytb phylogeny. Taking into account the lack of phylogenetic structure and that branch lengths are extremely short in this clade (Fig 6), this is likely to be a case of taxonomic inflation (4); Cytb may be revealing ongoing gene flow among populations within a single species. This should be posed as a working hypothesis to be tested with the addition of independent markers.

The genus *Myotis* comprises more than 100 species that originated during the last 10–15 million years [64], representing one of the most successful mammalian radiations. Conflict among morphological classifications, mitochondrial and nuclear gene trees, as well as among individual nuclear loci have been reported [26, 65, 66], suggesting that the *Myotis* radiation may have undergone phenomena such as lineage sorting, reticulation, and introgression [65]. Morphological variation is often a poor indicator of species-level relationships among this genus [67]; instead, geography showed to be a better predictor of phylogenetic relationships

than morphology [67]. In a mitochondrial phylogeny of New World *Myotis*, Larsen et al. [26] identified multiple cryptic lineages in *M. albescens*, *M. riparius*, and *M. nigricans*. In a recent study based on full mitochondrial genomes and targeted sequencing of nuclear ultraconserved elements (UCEs) of primarily New World *Myotis*, Platt et al. [65] found high levels of topological conflict between nuclear and mtDNA data, and also among nuclear loci, suggesting that hybridization and lineage sorting have also shaped the evolutionary relationships of the genus.

In agreement with the above-mentioned studies, in our analysis, four *Myotis* species (*M. riparius*, *M. nigricans*, *M. albescens*, and *M. levis*) failed to achieve reciprocal monophyly and depict low levels of genetic differentiation. Museum vouchers identified as *M. albescens* and *M. levis* resulted intermixed together and along with *M. nigricans*, a finding compatible with incomplete lineage sorting and/or introgression (sources of incongruence 1 and 2). These two factors may act in combination and probably in addition to limitations in morphological identification (3) and taxonomic errors (4) since in the presence of gene flow and/or low levels of genetic differentiation, morphological methods are more prone to errors in identification. The use of genomic approaches may help to differentiate which of the four mentioned sources of conflict are operating in *Myotis* species. Noteworthy, all *Myotis* field specimens fell within an exclusive *M. nigricans* clade, with low levels of divergence, which would belong to the same biological unit, and could have been erroneously grouped with other (paraphyletic) *M. nigricans* as a product of ecomorphological convergence that would in turn impact on taxonomy (sources of incongruence 3 and 4).

The alpha taxonomy of *Molossus*, although advanced during the past years, remains a work in progress. The genus is mainly Neotropical, occurring from the southeastern United States to southern Argentina. Conflict among morphological characters, acoustic patterns, mtDNA, as well as nuclear loci, has been reported in several studies [13, 28, 68]. Gager et al. [68] tested different sources of information for species discrimination in *Molossus* species. They found that although useful for separating particular species pairs, none of the applied methods (pelage coloration pattern, microsatellite analysis, mtDNA phylogeny, and geometric morphometrics) was infallible for the distinction of all recognized species. The high levels of conflict among different character sources are correlated with the low levels of genetic differentiation reported in this genus [13]. Recently, Loureiro et al. [9] published a high-resolution phylogeny of *Molossus*, based on single nucleotide polymorphisms (SNP) obtained by Genotyping by Sequencing (GBS), elevating the number of species in the genus from 11 to 14, revealing two cryptic species in *M. rufus*, and dividing a group with moderate levels of morphological differentiation into two species (*M. currentium* and *M. bondae*).

In our analysis, specimens 129 and 130, which were captured at the same locality, and were morphologically assigned to *M. molossus*, fell in different clades (Fig 8). This occurred with several specimen pairs captured at Santa Fe (samples 192 to 199), as well as with specimens SP27 and SP28, from Sao Paulo (Brazil), originally analyzed by Carnieli et al. [27]. This is expectable within morphologically conservative species with low genetic variation, and as was demonstrated, even multilocus approaches failed in species delimitation [13]. Only when an analysis based on thousands of SNPs was carried out, a well-supported phylogeny accounting for the distinction of independent evolutionary lineages within *Molossus* could be obtained [9].

## Conclusions

The approach applied in the present study confirms the general applicability of Cytb-based phylogenies in taxonomic identification of bats in Argentina at infrageneric and, partially, at specific levels. The suitability of this mtDNA marker has been recently validated in Europe in a qPCR protocol for species determination [29]. This is an important outcome since, in most

ecological and eco-epidemiological studies, the objective is to capture and release animals after taking small biopsies for their identification. In this context, where there is a deliberate impossibility of sacrificing and collecting each individual, Cytb phylogenetic identification is crucial to minimize possible external morphology-based identification errors, given that specimens cannot be reanalyzed. As mentioned, Cytb is useful for species-level delimitation in non-conflicting genera (*Eumops*, *Dasypterus*, *Molossops*) and has subgeneric resolution in more complex species groups (*Eptesicus*, *Myotis*, *Molossus*). Molecular processes (low genetic differentiation, incomplete lineage sorting), biological events (introgression), limitations in morphological identification, and errors in current taxonomy, may act in isolation or combination, having more impact in taxonomically challenging groups. In these cases, the use of a limited number of loci and/or the application of morphological approaches would be insufficient for species determination. The continuously increasing use and decreasing costs of genomic approaches (eg. GBS) may be the step forward to enhance accurate species delimitation and identification.

## Supporting information

**S1 File. Alignment including the three primers and multiple bat mtDNA genomes.** Primer sequences aligned with 19 bat mitogenomes corresponding to genera occurring in America (*Eptesicus*, *Lasiurus*, *Myotis*, *Tadarida*, *Artibeus*, and *Diaemus*). The region targeting primer Bat 05A was more conserved in all retrieved sequences compared to Bat 14A which showed internal gaps. The primer was Bat-Ep (this study) was designed to replace Bat 14A based on a conserved region.
(BIO)

**S2 File. List of 142 sequences retrieved from Genbank.** Members of all genera of the species that have verified distributions in the capture zone, including also allied species aiming to achieve the widest possible representation of the geographical range of those species, with special emphasis on South American species, were included.
(XLSX)

**S3 File. Matrix alignment 202 Cytb nucleotide sequences.** Alignment of 202 Cytb nucleotide sequences representing 41 species and 14 genera. No deletions, insertions or stop codons were either observed in the 1140 bp alignment.
(FAS)

**S1 Fig. Agarose gel 1.2% of PCR products obtained with primers Bat05A and Bat-Ep.** The combination of primers Bat 05A and Bat-Ep successfully amplified the Cytb gene and flanking regions in all assayed species, yielding a PCR product of ca 1330 bp. From left to right, PCR products were obtained in *Eptesicus* (lanes 1–5), *Dasypterus* (lane 6), *Molossops* (lane 7), *Molossus* (lane 8), *Myotis* (lane 9), and *Eumops* (lane 10). Lane 11 shows the PCR negative control, while lane 12 corresponds to the molecular-weight size marker.
(JPG)

**S1 Raw image. Raw file corresponding to S1 Fig.** Original and uncropped image underlying gel results.
(PDF)

## Acknowledgments

The authors would like to thank Sociedad Rural "Las Colonias", Sociedad Rural de Santa Fe, Municipalidad de Esperanza, Municipalidad de Recreo, Municipalidad de Santa Fe, Maria Lili

Dalla Fontana, Agustín Lastra and Fernando Carmona. DAC would like to thank Sabrina L. López, for her kind support.

## Author Contributions

**Conceptualization:** Diego A. Caraballo, María E. Montani, Valeria C. Colombo.

**Data curation:** María E. Montani.

**Investigation:** Diego A. Caraballo.

**Methodology:** Diego A. Caraballo, María E. Montani, Leila M. Martínez, Leandro R. Antoniazzi, Tomás C. Sambrana, Camilo Fernández, Valeria C. Colombo.

**Project administration:** Diego A. Caraballo, Daniel M. Cisterna, Fernando J. Beltrán.

**Resources:** Daniel M. Cisterna.

**Supervision:** Diego A. Caraballo.

**Validation:** Diego A. Caraballo.

**Visualization:** Diego A. Caraballo.

**Writing – original draft:** Diego A. Caraballo.

**Writing – review & editing:** Diego A. Caraballo, María E. Montani, Valeria C. Colombo.

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
