## [Decision Letter · Decision Letter 0]

2 Nov 2020

PONE-D-20-26748

Heterogeneous taxonomic resolution of cytochrome b gene identification of bats from Argentina: implications for field studies.

PLOS ONE

Dear Dr. Caraballo,

Thank you for submitting your manuscript to PLOS ONE. After careful consideration, we feel that it has merit but does not fully meet PLOS ONE’s publication criteria as it currently stands. Therefore, we invite you to submit a revised version of the manuscript that addresses the points raised during the review process.

We look forward to receiving your revised manuscript.

Kind regards,

Bi-Song Yue, Ph.D

Academic Editor

PLOS ONE

3. We note that Figure 1 in your submission contain map images which may be copyrighted. All PLOS content is published under the Creative Commons Attribution License (CC BY 4.0), which means that the manuscript, images, and Supporting Information files will be freely available online, and any third party is permitted to access, download, copy, distribute, and use these materials in any way, even commercially, with proper attribution. For these reasons, we cannot publish previously copyrighted maps or satellite images created using proprietary data, such as Google software (Google Maps, Street View, and Earth). For more information, see our copyright guidelines: http://journals.plos.org/plosone/s/licenses-and-copyright.

3.1.    You may seek permission from the original copyright holder of Figure 1 to publish the content specifically under the CC BY 4.0 license. 

3.2.    If you are unable to obtain permission from the original copyright holder to publish these figures under the CC BY 4.0 license or if the copyright holder’s requirements are incompatible with the CC BY 4.0 license, please either i) remove the figure or ii) supply a replacement figure that complies with the CC BY 4.0 license. Please check copyright information on all replacement figures and update the figure caption with source information. If applicable, please specify in the figure caption text when a figure is similar but not identical to the original image and is therefore for illustrative purposes only.

Reviewers' comments:

Reviewer's Responses to Questions

**Comments to the Author**

1. Is the manuscript technically sound, and do the data support the conclusions?

Reviewer #1: Partly

Reviewer #2: Yes

2. Has the statistical analysis been performed appropriately and rigorously? 

Reviewer #1: N/A

Reviewer #2: Yes

3. Have the authors made all data underlying the findings in their manuscript fully available?

Reviewer #1: Yes

Reviewer #2: Yes

4. Is the manuscript presented in an intelligible fashion and written in standard English?

Reviewer #1: No

Reviewer #2: Yes

5. Review Comments to the Author

Reviewer #1: This manuscript discusses the potential use of cytb mitochondrial sequences to identify bat species, an important task given that morphological identification is challenging. It’s a valid study providing some interesting data. I think the manuscript has potential but there are several issues that I think need to be addressed, some quite fundamental.

The manuscript seems rather hastily prepared. It contains many grammar/language issues and is not well organized. For examples figures are referred to either as “Fig” or “Fig.”, figure legends are therefore hard to find (and are not in a single sequence anywhere), figures are not presented in the correct order either which is confusing (for example, figure 1 is last). Great care should be taken in the revision of this manuscript to address these issues

More than that, fundamental issues, exemplified below by direct quotes, need to be considered in greater detail and with deeper scrutiny

“Incongruence between morphological and molecular identification may represent a methodological problem or a biological problem” True, but this statement totally fails to acknowledge two additional potential sources of incongruence: 1) errors in your own identification, and 2) errors in the existing taxonomy. Some such issues are raised and discussed in the Discussion section. However, the impression one gets is that existing taxonomy is taken ‘as given’ – fundamentally assumed to be correct, and the results are then judged based on the recovery of this taxonomy. This is obviously problematic. It is abundantly clear that bat taxonomy is very challenging (hence the difficulty in identifying species based on morphology), and a lot of it has been done without reference to molecular data. It is fair to assume that the current taxonomy is far from perfect (only in part reflects biological species), and hence some of your results may simply be reflecting that. The big question then is, can you tell in which case of incongruence which cause is at fault? That may be very challenging, and suggest that your overall findings may be far less clear than suggested in your Discussion and Conclusions. If you go case by case, are there clear examples of incongruences that are clearly caused by problems 1) molecular processes, 2) biology, 3) limitations of your identification, 4) errors in the current taxonomy? If you can in an organized manner identify and discuss each of these separately, I think it would be a lot easier for the reader to understand your results and their implications. If you cannot, it seems to me that your conclusions are more reflective of your opinion (treating taxonomy almost as a given), and thus less informative. Also, inevitably, as morphological identification becomes harder and therefore more error prone, inconsistencies between ID’s and any kind of data such as molecules, rises. This is a kind of a tautology, you need molecules to help you ID but to be able to do so you must first ID without molecules! Further discussion is needed. Example of problem 3) and/or 4) could be e.g. in E. bonariensis and patagonicus (Fig. 3). Dasypterus-Lasiurus (Fig. 4) seems to be a clear example of problem 4). In some cases it may be very hard to pinpoint the source of problem e.g. Fig. 7 and 8 – Could be 1), 2), 3), 4) or any combination thereof. Can you point to problems that clearly are the consequence of problem 1) or 2)?

“Morphological traits exhibit greater convergence and homoplasy compared with molecular data” I do not think you can make such a statement without a series of caveats. Just citing a study is not enough. This is clearly context dependent. Morphology is not a ‘thing’ that behaves equally across taxa or parts of organisms. Neither is rate of change and homoplasy even remotely constant across molecular data. There are many morophological traits of organisms that are extremely highly conserved and exhibit very low homoplasy (e.g. all mammals have mammary glands and produce milk, one origin, no loss; all spiders have spinnerets and produce silk, single origin and no loss of spinnerets or certain silk glands – there are countless examples) and there are some extremely highly homoplastic DNA sequences, the D-loop is one example, but they are also countless. What are you trying to convey with this statement

“Another one, more applicable for mitochondrial genes, is the high rate of evolution, which might cause homoplasy [54,55]” Homoplasy is also, of course, in itself information that can aid in the resolution of phylognies! J. Wenzel sometime famously claimed “homoplasy is your friend”. Again, can you point directly to cases where homoplasy IS misleading?

“The yellow bats of the genus Dasypterus were monophyletic in the Cytb analysis but were 351 paraphyletic with respect to Lasiurus” This statement is not accurate. Dasypterus is monophyletic, period. However, Lasiurus is not, unless it were to include Dasypterus. Another way of stating that is that the results imply that Dasypterus ega should be transferred to Lasiurus (pending further support..) This seems like a clear example of errors in the existing taxonomy, that you could link with discussion re the four sources of incongruence outlined above.

“Beyond any possible error in morphological identification, such low genetic distances are prone to produce non reciprocally monophyletic species clades” Perhaps because taxonomists have not accurately identified biological species! In other words, the taxonomy may simply be in error, and the genetics are revealing ongoing geneflow as expected among populations within a species. How do you know if the problem is molecular processes (incomplete lineage sorting etc) or erroneous taxonomy?

On a different issue. I am missing some discussion on the use of DNA ‘barcodes’ in general. Your approach is not exactly barcoding, and certainly does not use all the typical tools of that field. There is a lot of discussion in barcoding literature on ID versus Monophyly. These are certainly not the same things. At least some proponents of the Barcoding movement are much more concerned with the ability to identify specimens, than phylogenetic reconstruction and monophyly… and the end goal of barcoding is never—per se—a novel phylogeny (for which, nowadays, we hope to have more data).

In sum. I think this study presents interesting data and reveals challenging problems However, it would be made much more valuable if the results were discussed more clearly and precisely in the context of each of the “problems 1-4”. An attempt should be made to allocate to each of the instances of incongruence to each of these problems – and those that cannot be allocated thusly, should be identified as potentially the result of any of these in isolation or combination. This would be a way to acknowledge better the complexity of the problem, and the reality that it is hard to pinpoint the exact cause(s) of incongruence, especially in taxonomically challenging groups. You may also discuss how to move forward given these issues…

Sincerely, Ingi Agnarsson

Reviewer #2: PONE-D-20-26748, "Heterogeneous taxonomic resolution of cytochrome b gene identification of bats from Argentina: implications for field studies."

This manuscript addresses the use of cyt b gene sequence data for identification of bat species in the province of Santa Fe, Argentina. Species identifications based on sequence data were compared to those based on morphology. Their results supported the general applicability of cyt b based phylogenies in eco-epidemiological studies where nonlethal sampling is required. However, resolution varied greatly depending on genetic differentiation with the genera of interest.

The study represents a good case study on the utility of cytb to provide identifications of problematic taxa in the field. The increase in non-lethal sampling projects globally necessitates an understanding of the potential error rates in species identifications in these types of projects. Improper identification of pathogen hosts has direct negative implications on our understanding of emerging infectious diseases and hampers mitigation efforts. This issue has been well illustrated by our current lack of understanding in regard to the origins and reservoir species of the novel corona virus. The highly diverse order Chiroptera is certainly a good example of a need for robust methods of identification, but this holds for all taxa associated with pathogens of interest.

Line 74 – while it is true that the material collected by a wing punch can provide DNA for a small number of specific studies, the value to future research is greatly hampered by sample size and breadth. Future studies of viral pathogens, parasites, isotopic analyses, etc. are not possible. Samples are always going to be the limiting factor as new questions arise and new technology is developed. Might be nice to emphasize this.

Line 87-91– agreed that Cyt b provides better resolution and there is more cytb data in GenBank for comparative analyses. Might be worth noting cases where single gene taxonomic hypotheses were revised based on multigene (including nuclear) analyses.

Line 95 – As this manuscript derives its data from a pathogen surveillance program and we are in the midst of a pandemic it would be topical to provide some tie in to the current situation and collection based biodiversity sampling for understanding pathogens. For example:

DiEuliis et al. 2016. Opinion: Specimen collections should have a much bigger role in infectious disease research and response. Proceedings of the National Academy of Sciences 113: 4–7.

Cook et al, 2020. Integrating Biodiversity Infrastructure into Pathogen Discovery and Mitigation of Emerging Infectious Diseases. BioScience, 70(7), pp.531-534.

Line 121-124 – commend the authors on also collecting full voucher specimens as these enable a wide range of further research that is not possible with mark-release programs. Any studies of viral pathogens, parasites, isotopic analyses require full vouchers with high quality tissues, and any assessment of environmental change requires archived museum vouchers as baselines.

Footnotes for table 1 and 2 – spelling of “reciprocally”

Table 2 – Inclusion of museum catalog numbers provides critical ability to retest these hypotheses in subsequent research.

Footnote of Table 2 – spelling of “georeference”

Line 146-Primer development and sequencing protocols are appropriate

Line 180 – unable to locate Genbank Accession Numbers (MT262814 - MT262873) in NCBI

Line 311 – Gold standard for GenBank is sequence associated with vouchered specimen

Suggest not focusing as much on specific phylogenetic findings in the chiropteran phylogenies because taxon sampling and single gene analyses are an issue. They do represent good examples for a proof of concept illustrating the potential short falls of using cytb alone garner precise identifications.

Recent paper on Myotis systematics may be useful. Carrion-Bonilla, C.A. and Cook, J.A., 2020. A new bat species of the genus Myotis with comments on the phylogenetic placement of M. keaysi and M. pilosatibialis. THERYA, 11(3), p.508-532.

Clearly, addition of any molecular data to mark-release projects provides added rigor. However, as the study illustrates, Cytb alone is not sufficient to give unequivocal species identifications. In studies of pathogens or parasites it is critical to have a robust understanding of host ecology and thus knowing the precise identification of the host taxon is essential.

The drawback of these types of studies is that proper identifications are dependent on the quality and breadth of sequence data available for comparative analyses as well as the issues associated with mitochondrial capture and introgression pointed out by the authors.

The authors do a good job of addressing the potential shortcomings of these types of analyses and the reasons behind them. It would be useful if the paper also outlined the pros and cons of mark-release vs lethal collection and deposition in collections in order to build biological infrastructure for future research on EIDs and environmental change.

The study represents a useful contribution in that it allows for an assessment of utilizing cytb sequence data for providing identifications of the Argentinian bat fauna and the findings can be generalized in terms of using this methodology in similar mark-release programs elsewhere.

6. PLOS authors have the option to publish the peer review history of their article (what does this mean?). If published, this will include your full peer review and any attached files.

Reviewer #1: **Yes: **Ingi Agnarsson

Reviewer #2: No

---

## [Author Response · Author response to Decision Letter 0]

13 Dec 2020

Response to Academic Editor

1. We have attended to PLOS ONE’s style requirements as requested.

2. We provide a pdf file with the raw image of S1 Fig (agarose gel) also as Supporting Information, according to the journal’s guidelines.

3. We have modified Figure 1, using one of the recommended resources (Natural Earth (public domain): http://www.naturalearthdata.com/)

Response to Reviewer #1

The manuscript seems rather hastily prepared. It contains many grammar/language issues and is not well organized. For examples figures are referred to either as “Fig” or “Fig.”, figure legends are therefore hard to find (and are not in a single sequence anywhere), figures are not presented in the correct order either which is confusing (for example, figure 1 is last).

RESPONSE: We thank the Reviewer’s comments and suggestions. We have named and cited Figures, Tables and Supporting Information following the journal’s guidelines. We have also checked the order in which these are presented in the pdf version. We have reviewed grammar and made a considerable effort to improve the manuscript’s organization.

“Incongruence between morphological and molecular identification may represent a methodological problem or a biological problem” True, but this statement totally fails to acknowledge two additional potential sources of incongruence: 1) errors in your own identification, and 2) errors in the existing taxonomy. Some such issues are raised and discussed in the Discussion section. However, the impression one gets is that existing taxonomy is taken ‘as given’ – fundamentally assumed to be correct, and the results are then judged based on the recovery of this taxonomy. This is obviously problematic. It is abundantly clear that bat taxonomy is very challenging (hence the difficulty in identifying species based on morphology), and a lot of it has been done without reference to molecular data. It is fair to assume that the current taxonomy is far from perfect (only in part reflects biological species), and hence some of your results may simply be reflecting that. The big question then is, can you tell in which case of incongruence which cause is at fault? That may be very challenging, and suggest that your overall findings may be far less clear than suggested in your Discussion and Conclusions. If you go case by case, are there clear examples of incongruences that are clearly caused by problems 1) molecular processes, 2) biology, 3) limitations of your identification, 4) errors in the current taxonomy? If you can in an organized manner identify and discuss each of these separately, I think it would be a lot easier for the reader to understand your results and their implications. If you cannot, it seems to me that your conclusions are more reflective of your opinion (treating taxonomy almost as a given), and thus less informative. Also, inevitably, as morphological identification becomes harder and therefore more error prone, inconsistencies between ID’s and any kind of data such as molecules, rises. This is a kind of a tautology, you need molecules to help you ID but to be able to do so you must first ID without molecules! Further discussion is needed. Example of problem 3) and/or 4) could be e.g. in E. bonariensis and patagonicus (Fig. 3). Dasypterus-Lasiurus (Fig. 4) seems to be a clear example of problem 4). In some cases it may be very hard to pinpoint the source of problem e.g. Fig. 7 and 8 – Could be 1), 2), 3), 4) or any combination thereof. Can you point to problems that clearly are the consequence of problem 1) or 2)?

RESPONSE: We have taken the Reviewer’s recommendation and restructured the manuscript focusing on the discussion of problems 1-4. This decision implied major modifications throughout the manuscript, and although the main findings remain unchanged, the discussion of results was centered in the four causes of conflict, as suggested by the Reviewer.

“Morphological traits exhibit greater convergence and homoplasy compared with molecular data” I do not think you can make such a statement without a series of caveats. Just citing a study is not enough. This is clearly context dependent. Morphology is not a ‘thing’ that behaves equally across taxa or parts of organisms. Neither is rate of change and homoplasy even remotely constant across molecular data. There are many morophological traits of organisms that are extremely highly conserved and exhibit very low homoplasy (e.g. all mammals have mammary glands and produce milk, one origin, no loss; all spiders have spinnerets and produce silk, single origin and no loss of spinnerets or certain silk glands – there are countless examples) and there are some extremely highly homoplastic DNA sequences, the D-loop is one example, but they are also countless. What are you trying to convey with this statement

RESPONSE: We have reconsidered this statement after the Reviewer’s comment. We agree that homoplasy may act at both levels (morphological and molecular) and in varying degrees depending on which character and/or gene (or even codon position) are taken into consideration.

“Another one, more applicable for mitochondrial genes, is the high rate of evolution, which might cause homoplasy [54,55]” Homoplasy is also, of course, in itself information that can aid in the resolution of phylognies! J. Wenzel sometime famously claimed “homoplasy is your friend”. Again, can you point directly to cases where homoplasy IS misleading?

We have reconsidered this statement after the Reviewer’s criticism. As in the previous point, we agree that homoplasy is not necessarily totally misleading, and may thus be at least partially informative.

“The yellow bats of the genus Dasypterus were monophyletic in the Cytb analysis but were 351 paraphyletic with respect to Lasiurus” This statement is not accurate. Dasypterus is monophyletic, period. However, Lasiurus is not, unless it were to include Dasypterus. Another way of stating that is that the results imply that Dasypterus ega should be transferred to Lasiurus (pending further support..) This seems like a clear example of errors in the existing taxonomy, that you could link with discussion re the four sources of incongruence outlined above.

RESPONSE: We modified a misinterpretation: The species Dasypterus xanthinus (Baird et al. 2015, 2017) is annotated in Genbank as member of the genus Lasiurus. Both genera are reciprocally monophyletic based on multi-locus analyses, so we re-interpreted this result, owing that Cytb does not conflict with current taxonomy. We have modified Figs 2 and 4 according to this reinterpretation.

Baird AB, Braun JK, Mares MA, Morales JC, Patton JC, Tran CQ, et al. Molecular systematic revision of tree bats (Lasiurini): doubling the native mammals of the Hawaiian Islands. J Mammal. 2015;96: 1255–1274. doi:10.1093/jmammal/gyv135

Baird AB, Braun JK, Engstrom MD, Holbert AC, Huerta MG, Lim BK, et al. Nuclear and mtDNA phylogenetic analyses clarify the evolutionary history of two species of native Hawaiian bats and the taxonomy of Lasiurini (Mammalia: Chiroptera). Etges WJ, editor. PLoS One. 2017;12: e0186085. doi:10.1371/journal.pone.0186085 

“Beyond any possible error in morphological identification, such low genetic distances are prone to produce non reciprocally monophyletic species clades” Perhaps because taxonomists have not accurately identified biological species! In other words, the taxonomy may simply be in error, and the genetics are revealing ongoing geneflow as expected among populations within a species. How do you know if the problem is molecular processes (incomplete lineage sorting etc) or erroneous taxonomy?

RESPONSE: This issue is addressed in the Discussion section.

On a different issue. I am missing some discussion on the use of DNA ‘barcodes’ in general. Your approach is not exactly barcoding, and certainly does not use all the typical tools of that field. There is a lot of discussion in barcoding literature on ID versus Monophyly. These are certainly not the same things. At least some proponents of the Barcoding movement are much more concerned with the ability to identify specimens, than phylogenetic reconstruction and monophyly… and the end goal of barcoding is never—per se—a novel phylogeny (for which, nowadays, we hope to have more data).

RESPONSE: We have removed the discussion about the use of barcodes in the introduction. We agree that it might open a debate that is not retaken along the paper.

In sum. I think this study presents interesting data and reveals challenging problems However, it would be made much more valuable if the results were discussed more clearly and precisely in the context of each of the “problems 1-4”. An attempt should be made to allocate to each of the instances of incongruence to each of these problems – and those that cannot be allocated thusly, should be identified as potentially the result of any of these in isolation or combination. This would be a way to acknowledge better the complexity of the problem, and the reality that it is hard to pinpoint the exact cause(s) of incongruence, especially in taxonomically challenging groups. You may also discuss how to move forward given these issues…

RESPONSE: As mentioned above, we have restructured the manuscript focusing in the four problems suggested by the Reviewer. We have identified strengths and limitations of the methodology applied in this study, and mentioned the alternatives to overcome such limitations.

Response to Reviewer #2

Line 74 – while it is true that the material collected by a wing punch can provide DNA for a small number of specific studies, the value to future research is greatly hampered by sample size and breadth. Future studies of viral pathogens, parasites, isotopic analyses, etc. are not possible. Samples are always going to be the limiting factor as new questions arise and new technology is developed. Might be nice to emphasize this.

RESPONSE: We agree with the reviewer's observation, and have emphasized the limitations linked to wing punch tissue collection.

Line 87-91– agreed that Cyt b provides better resolution and there is more cytb data in GenBank for comparative analyses. Might be worth noting cases where single gene taxonomic hypotheses were revised based on multigene (including nuclear) analyses.

RESPONSE: We attended this issue in the discussion. We think that otherwise it would enlarge the Introduction.

Line 95 – As this manuscript derives its data from a pathogen surveillance program and we are in the midst of a pandemic it would be topical to provide some tie in to the current situation and collection based biodiversity sampling for understanding pathogens. For example:

DiEuliis et al. 2016. Opinion: Specimen collections should have a much bigger role in infectious disease research and response. Proceedings of the National Academy of Sciences 113: 4–7.

Cook et al, 2020. Integrating Biodiversity Infrastructure into Pathogen Discovery and Mitigation of Emerging Infectious Diseases. BioScience, 70(7), pp.531-534.

RESPONSE: We thank the Reviewer's suggestion. We have included a paragraph pointing out the value of scientific collections in pathogen surveillance.

Line 121-124 – commend the authors on also collecting full voucher specimens as these enable a wide range of further research that is not possible with mark-release programs. Any studies of viral pathogens, parasites, isotopic analyses require full vouchers with high quality tissues, and any assessment of environmental change requires archived museum vouchers as baselines.

RESPONSE: We thank the Reviewers recommendation. We agree in that full voucher specimen collection has an incomparable potential. In this study we collected 6 specimens that were taxonomically interesting. We prepared skin, skull, skeleton and tissues for DNA extraction. At the present we are collecting full voucher specimens in an ongoing project involving viral pathogens, parasites and using Cytb for species identification in bats, including also the preparation of soft organs and cryopreserved tissues for RNA extraction.

Footnotes for table 1 and 2 – spelling of “reciprocally”

RESPONSE: Corrected. Thanks for the observation.

Table 2 – Inclusion of museum catalog numbers provides critical ability to retest these hypotheses in subsequent research.

Footnote of Table 2 – spelling of “georeference”

RESPONSE: Corrected. Thanks for the observation.

Line 146-Primer development and sequencing protocols are appropriate

Line 180 – unable to locate Genbank Accession Numbers (MT262814 - MT262873) in NCBI

RESPONSE: We have requested that data are to be held confidential until Apr 1, 2021. They will not be released to the public database until this date, or until the data or accession numbers appear in print, whichever is first.

Line 311 – Gold standard for GenBank is sequence associated with vouchered specimen

Suggest not focusing as much on specific phylogenetic findings in the chiropteran phylogenies because taxon sampling and single gene analyses are an issue. They do represent good examples for a proof of concept illustrating the potential short falls of using cytb alone garner precise identifications.

RESPONSE: We thank the Reviewer’s suggestion. In concordance with Reviewer’s 1 suggestion, we have focused our discussion in four sources of incongruence that may act separately or in combination: 1) molecular processes, 2) biology, 3) limitations in identification, and 4) errors in the current taxonomy.

Recent paper on Myotis systematics may be useful. Carrion-Bonilla, C.A. and Cook, J.A., 2020. A new bat species of the genus Myotis with comments on the phylogenetic placement of M. keaysi and M. pilosatibialis. THERYA, 11(3), p.508-532.

RESPONSE: We have included the referred paper as an example of the continuous revision of the systematics of Myotis. 

Clearly, addition of any molecular data to mark-release projects provides added rigor. However, as the study illustrates, Cytb alone is not sufficient to give unequivocal species identifications. In studies of pathogens or parasites it is critical to have a robust understanding of host ecology and thus knowing the precise identification of the host taxon is essential.

RESPONSE: We agree with the Reviewer. We underlined the scopes and limits of the use of Cytb in bat species identification, and pointed out that genomic approaches would be needed especially in taxonomically challenging lineages.

The drawback of these types of studies is that proper identifications are dependent on the quality and breadth of sequence data available for comparative analyses as well as the issues associated with mitochondrial capture and introgression pointed out by the authors.

The authors do a good job of addressing the potential shortcomings of these types of analyses and the reasons behind them.

It would be useful if the paper also outlined the pros and cons of mark-release vs lethal collection and deposition in collections in order to build biological infrastructure for future research on EIDs and environmental change.

RESPONSE: We have addressed this topic in the Introduction.

The study represents a useful contribution in that it allows for an assessment of utilizing cytb sequence data for providing identifications of the Argentinian bat fauna and the findings can be generalized in terms of using this methodology in similar mark-release programs elsewhere.

---

## [Editor Report · Decision Letter 1]

16 Dec 2020

Heterogeneous taxonomic resolution of cytochrome b gene identification of bats from Argentina: implications for field studies.

PONE-D-20-26748R1

Dear Dr. Caraballo,

We’re pleased to inform you that your manuscript has been judged scientifically suitable for publication and will be formally accepted for publication once it meets all outstanding technical requirements.

Kind regards,

Bi-Song Yue, Ph.D

Academic Editor

PLOS ONE

---

## [Editor Report · Acceptance letter]

23 Dec 2020

PONE-D-20-26748R1 

Heterogeneous taxonomic resolution of cytochrome b gene identification of bats from Argentina: implications for field studies. 

Dear Dr. Caraballo:

I'm pleased to inform you that your manuscript has been deemed suitable for publication in PLOS ONE. Congratulations! Your manuscript is now with our production department. 

Kind regards, 

on behalf of

Dr. Bi-Song Yue 

Academic Editor

PLOS ONE